# Information retrieval interfaces in virtual reality—A scoping review focused on current generation technology

Maurice Schleußinger ⬤ *

Department of Information Science, Heinrich Heine University, Düsseldorf, Germany

* maurice.schleussinger@hhu.de

**Data Availability Statement:** All relevant data are within the manuscript and its Supporting information files as well as the Zenodo data repository: https://doi.org/10.5281/zenodo.3985949.

## Abstract

The Information Retrieval user experience has remained largely unchanged since its inception for computers and mobile devices alike. However, recent developments in Virtual Reality hardware (pioneered by *Oculus Rift* in 2013) could introduce a new environment for Information Retrieval. This paper reports the results of a Scoping Literature Review (PRISMA-ScR) by rigorously examining the entire body of relevant literature with reproducible methods. The following research questions are answered: "What prototypes and concepts of Virtual Reality Information Retrieval systems with current generation hardware exist?", "How are user interaction and especially user input realised in these systems?", "What Retrieval features are used in these systems?", "How are search results displayed in these systems?" and "Can these VR IR systems compare to traditional (non-VR) IR systems?". After querying Google Scholar, Scopus and Web of Science, 1042 documents were reviewed in depth. Key features and attributes of the systems are summarised and discussed. Sketches of the user interfaces are included as well. The 30 documents that were relevant to the research questions include 16 distinct systems or theories. They discuss and utilise several user input technologies, ranging from controllers, voice input or hand tracking. Although conventional retrieval features are less common, systems enable retrieval of literature, 3D objects, images, books and texts and arrange them in a virtual space (e.g. as grids, arcs or maps). Finally, many of these systems were compared to conventional counterparts through user evaluation (n = 10). Most found user task times to be shorter or equal (n = 5, n = 3). In the seven papers that measured user performance (rate of correct solutions), three reported better performance (one equal). Notably, users always were more satisfied with the Virtual Reality systems compared to conventional ones. Possible limitations of these evaluations are demographic selection and the quality of baseline systems (control).

## 1 Introduction

### 1.1 Rationale

Computers made their way from being room-filling machines with restricted interaction capabilities to powerful and mobile systems with graphical user interfaces. Since then they also became the preferred way to store and retrieve digital information.

**Funding:** This study was funded by Heinrich Heine University Düsseldorf.

**Competing interests:** The authors have declared that no competing interests exist.

Search interfaces enable users to easily retrieve locally or remotely saved information with little effort. With the popularisation of internet search engines (for example *Baidu* and *Google*) and local file search (for example *Microsoft Windows start menu search* and *Apple macOS Spotlight search*) also came the simplification from specialised query languages (with logical operators and special characters) to natural language input with keywords or entire phrases or questions. Finally, the popularisation of smartphones and mobile internet coverage brought the ability to find virtually any digital information online within seconds to the average citizen.

Speech recognition aside, the general user experience for search engines changed little ever since the very first retrieval systems were developed. A person enters the search query with a keyboard (a virtual one for smartphones) into a search bar. The search results are displayed in a list of representative snippets ranked by their relevance to the input query. But, a major development in Information Technology might hold the potential for new paradigms regarding search: The significantly improved hardware for Virtual Reality experiences of recent years.

Similar to the leap in innovation from mobile phones to smartphones, the release of the first working prototypes of the *Oculus Rift* VR headset marked a major change in both the quality and availability of Virtual Reality (VR) hardware. The new generation of VR headsets is intended to be usable by everybody at their homes without any technical background or training. Some do not require a separate computer or gaming console, some do not require any external sensors at all. In contrast to prior systems, they are less expensive and provide a more immersive experience ([1]). Current generation systems aim to be easy to use and provide a pleasant experience for the user by having adequate per-eye display resolution, high frame-rates, and precise sensor tracking. As [2] puts it, "[. . .] the advent of modern head-mounted displays marks a new paradigm for VR". This development also invites new research which explores the possibilities of this new generation of hardware. These systems typically include the head-mounted display itself, some kind of input device and sometimes dedicated devices to enable tracking of the display and input devices. Fig 1 shows the components of the HTC Vive VR system. Additionally, some systems include a build-in computer to process graphics of the virtual world, while others need to be connected to an external computer.

With the capabilities of this current generation hardware, more and more productive systems take advantage of Virtual Reality for use cases like surgery training, pain management, treatment of mental conditions [3], conferencing [4], or equipment and operational training [5] and more. Many of these systems also require efficient retrieval of specific information. At the very least, one frequently wants to be able to quickly search for relevant information in almost any professional context. Since the dawn of the World Wide Web Information Retrieval became essential as it "deals with the representation, storage, organisation of, and access to information items". Also, these items should "provide the user with easy access to the information in which he is interested." [6] These definitions still hold today. But, how should an Information Retrieval interface function and look like with current generation Virtual Reality systems?

While there is existing work (existing systems) in this specific area (including commercial software, such as in Fig 2), no comprehensive literature review was found, especially not one with a focus on the body of recent research in Interactive Information Retrieval systems in Virtual Reality. Such a review should be both rigorous and reproducible and provide an adequate and complete overview of said academic literature. However, *(literature) review* is the general term for several more or less well-defined activities conducted by researchers or students to survey the body of existing research literature. These activities can range from a chapter in a student's theses, or a paragraph in the first section of most research articles to individual works of research. As "researchers have used numerous terms to depict their review methods and

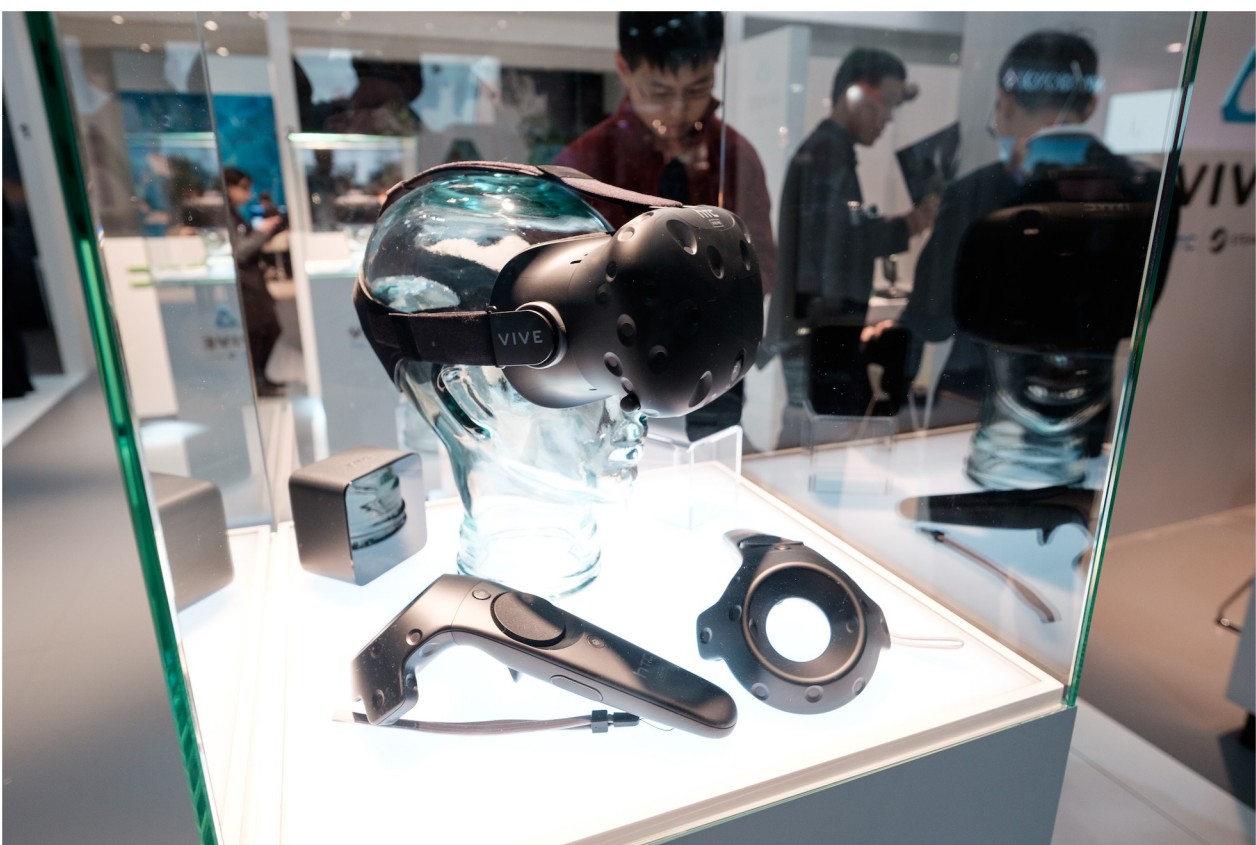

**Fig 1. HTC Vive Virtual Reality system with a head-mounted display (top center), tracking device (left) and input devices (bottom left and right).**

approaches", these include narrative, theoretical, critical, descriptive, comprehensive or systematic reviews or the meta-analysis [7]. Without going further into details about the uses of each of these review methods, it can be said that while there are valid use-cases for different kinds of reviews. But, as [8] summarise about systematic reviews: "if a review purports to be an authoritative summary of what "the evidence" says, then the reader is entitled to demand that this is a comprehensive, objective, and reliable overview, and not a partial review of a convenience sample of the author's favourite studies". These rigorous and protocol-based literature reviews are today most often seen in the fields of Medicine and Social Science. They supersede a general literature review by having concrete requirements of methodology. Most importantly, they aim to be reproducible and as objective as possible. While often conducted in certain fields, the requirements for these reviews are not inherently limited to these fields. As [9] report, "despite its indisputable benefits, [. . .] [systematic reviews] are yet infrequently used in Library and Information Science research [. . .] and we propose the adoption of the systematic review as a methodology for recovering, analysing, evaluating and critically appraising the relevant literature in library and information science [. . .]." Efforts to further standardise protocols for reviews include the detailed and widely accepted PRISMA guidelines [10]. These guidelines consist of 27 items to include in a systematic review or meta-analysis ranging from title and abstract to conclusions and funding. Each item tasks the author to report on central aspects of their review process, such as the eligibility criteria, search strategy, synthesis methods and so on. It also includes a template for a flow chart that showcases how many records

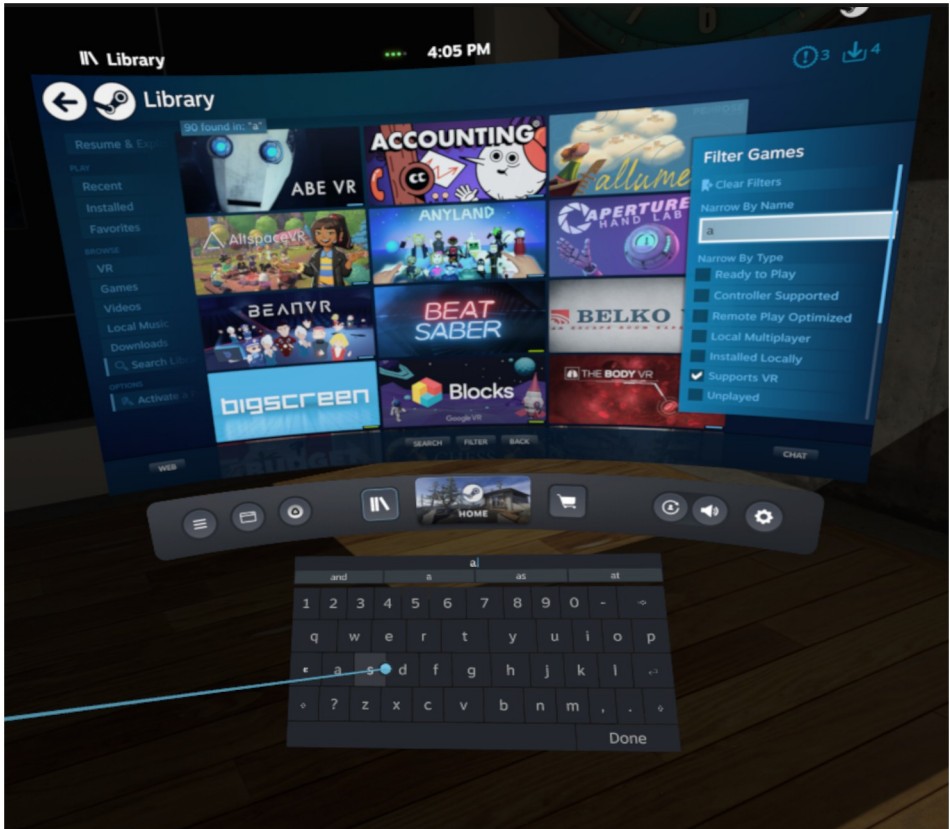

**Fig 2. Information Retrieval functionality on the SteamVR platform with a flat result list and virtual keyboard.**

were initially collected and which records were excluded at each step of the synthesis for what reasons. Given the empirical nature and the probing design of the research questions of this review, a structured and protocol-based review is sensible. However, the given study reports the results of a scoping review instead of a systematic review. Scoping reviews are similar to systematic reviews but differ in their somewhat broader goal to "identify knowledge gaps, scope a body of literature, clarify concepts, investigate research conduct, or to inform a systematic review" in contrast to the typically more narrow research question(s) of a systematic review. Most fittingly, scoping reviews can be used to "investigate the design and conduct of research on a particular topic" [11], the encompassing goal of the given research questions. The given review is based on the extended PRISMA protocol for scoping reviews known as PRISMA-ScR [12]. The PRISMA-ScR guidelines include a total of 20 obligatory and 2 optional items to report on, all based on the aforementioned PRISMA guidelines for systematic reviews. The item list was pruned by an expert panel to better fit the format of a scoping review [12]. For example, in a PRISMA-ScR review the critical appraisal of the identified literature is optional and the synthesis can be more compact (for example in tabular format and without summary measures or additional analyses).

## 1.2 Objectives

This scoping review analyses recent work on Interactive Information Retrieval systems and answers the following research questions:

- *RQ 1*: What prototypes and concepts of Virtual Reality Information Retrieval systems with current generation hardware exist?

- *RQ 2*: How are user interaction and especially user input realised in these systems?

- *RQ 3*: What retrieval features are used in these systems?

- *RQ 4*: How are search results displayed in these systems?

- *RQ 5*: Can these VR IR systems compare to traditional (non-VR) IR systems?

To be considered for this review, research should utilise recent developments in consumer-grade VR hardware, starting with the *Oculus Rift* DK1 release on March 2013, as the availability of this device marked the start of releases of current generation systems [1]. This includes, but is not limited to, VR headsets with high-resolution displays, precise tracking of the headset and some form of user input (gazing, controllers, hand tracking, etc.). Finally, a theory, a concept or prototype design of a search interface with result presentation should be the focus of the work. A limitation to the publication years 2013 and up is reasonable for theoretical work as well, since the limit ensures that authors have been (at least hypothetically) exposed to current-generation VR systems.

## 2 Methods

### 2.1 Protocol and registration

The given review is based on the extended PRISMA protocol for scoping reviews known as PRISMA-ScR [12] and also follows some of the more specific guidelines provided by [13] about Systematic Literature Reviews of Information Systems as appropriate. Here, the aforementioned PRISMA-ScR guidelines are still fully respected and implemented, but some even more specific guidelines are respected as well (for example practical screen and methodical screen).

### 2.2 Eligibility criteria

As suggested by [13], practical and methodical screens (filters) were defined before any literature was screened:

Practical screen:

- Topic: Virtual Reality AND Information Retrieval

- Language: English

- Year of publication: 2013 or later

- Journals & conferences: any

- Scientific: Only academic literature such as published theses or peer-reviewed papers

The practical screen only limits the broad topic, which has direct relevance for the research questions. It filters only publications written in English as it is the general norm as the primary publication language for scientific literature. The limit on the publication years also directly corresponds to the research questions. Finally, while generally only scientific publications are considered, student theses are also included. Even if non-peer-reviewed works could potentially be of poor quality or report unconfirmed or wrong results, this would be mentioned in this review (even though it doesn't provide a full critical appraisal) and could be evaluated by the literate reader of this review. On the other hand, student theses may also provide more

detail about a system or theory that is introduced in peer-reviewed work by the same or similar authors.

**2.2.1 Augmented vs. Virtual Reality.** Simply put, Virtual Reality uses "images and sounds created by a computer that seem almost real to the user, who can interact with them by using sensors" [14], while Augmented Reality is "a technology that combines computer-generated images on a screen with the real object or scene that you are looking at" [15]. As the design philosophy and available input and output can therefore vary vastly, Augmented Reality systems are not the focus of this review. Any literature about Augmented Reality is already implicitly excluded in the practical screen. However, sampling database searches conducted before to this review revealed a tendency in the body of literature to mention Virtual Reality very briefly, while focusing on systems or concepts about Augmented Reality. These works typically included the term Augmented Reality in the title or topic. By excluding this term in the database queries whenever possible, the number of works focused on Augmented Reality could be reduced. Furthermore, the first screening also excluded works with an obvious focus on Augmented Reality where the query could not exclude such works beforehand.

## 2.3 Information sources

As this review aims to assess activity in Virtual Reality research in general, it is important to search databases with very good coverage of the body of relevant work. Besides *Web of Science* ([16]) and *Scopus* ([17]), the "two major existing multidisciplinary databases", *Google Scholar* ([18]) is also included, as it achieves very general good coverage ([19, 20]). While there are many discipline-specific databases as well, these are primarily intended for practitioners of these fields. Today, general databases tend to provide access to a superset of research items. As [19] state: "[c]itation counts from a range of different sources have been shown to correlate positively with [Google Scholar] citation counts at various levels of aggregation". Nonetheless, such a theory should not be followed without some confirmation at least. Many of the works relevant to the research questions can be attributed to the field of computer science. As both ACM and IEEE are some of the biggest publishers in the field of computer science and also provide direct database access to their publications, their databases were also probed to test if they would yield additional relevant results. However, as both of these databases are also indexed by *Google Scholar*, no additional relevant works could be identified with probing database queries. Therefore, they were not included separately. Accordingly, no other relevant discipline-specific databases were identified. Finally, considering the limit to works from the year 2013 and above, all relevant literature can be expected to be available online. Therefore, no analogue search was conducted. The electronic database searches were executed between January 15th and February 5th 2020 for the years 2013 to 2019 and repeated September 18th 2020 for the year 2020 up until that date. Additionally, the reference lists of all relevant articles were screened for publications relevant to the research questions as well as all other publications of the first author of all relevant works. The same practical and methodical screens were applied here.

## 2.4 Search

One of the key challenges of a literature review is fine-tuning the electronic search strategy. One needs to formulate adequate queries for each database, which should expose all relevant works of research without increasing the overall workload of the review to an unmanageable level. For the given research questions, the overarching fields of research are Computer Science and Information Science, but works outside of these fields could also be relevant. Some would

categorise relevant works in the field of Human-Computer Interaction. And specifically, the relevant research areas are Virtual Reality and Information Retrieval.

Exploratory queries conducted prior and parallel to this review helped to identify common terms used in relevant works. For this review, relevant academic works would focus on a Virtual Reality system that provides some kind of search functionality, which is a central part of Information Retrieval systems.

The term *Virtual Reality* is well established in academic literature as well as the general public. The Dewey Decimal Classification defines Virtual Reality in "006.8 Augmented and virtual reality". The IEEE Taxonomy even lists separate keywords for *Virtual Reality* and *Augmented Reality*. Relevant conferences are named *Virtual Reality Software and Technology* or *IEEE Virtual Reality*. And even commercial systems are marketed as *Virtual*, *Augmented* or *Mixed Reality* systems by all major producers ([21–25]). There are some terms for Virtual Reality. For example, the Cave Automatic Virtual Environment ([26]), which was developed and improved before 2013 and is therefore not relevant for this review or the term *Virtual Environment* which is used almost exclusively in conjunction with the more common term *Virtual Reality*. Or the more hardware-oriented term *head mounted display* which technically describes just the Virtual Reality headset itself. To verify that this term is not also used prevalently, exploratory searches with variations of the query "head-mounted display" AND "information retrieval" (2013-2020) for the selected three databases were conducted. The search in Scopus lists 19 results, and the more common terms (MR, AR or VR) were either included in the title (n = 8), the abstract(n = 5) or as keywords (n = 3), while the remaining entries were not relevant to the research questions (n = 3). Google Scholar yields many results (n = 602). However, the first ten database results either use one of the more common terms (MR, AR or VR) in the title (n = 4), the abstract (n = 3) or are not relevant at all (n = 3). Finally, Web of Science lists a total of nine results, none of which are relevant for the research questions.

The term *Information Retrieval* is widely used as well, but may also include many aspects that are not relevant for this review (e.g. performance measurements with established datasets). Additionally, at least in theory, one could produce a system with search functionality without knowledge of the term or the field. But even then, the term *search* should be mentioned at some point. Thereby the following queries were used for the 3 relevant databases:

- Query for *Google Scholar*: *interface AND search AND system AND "Information Retrieval" AND "Virtual Reality" Publication years: 2013-2020*

- Query for *Web of Science*: *(TS=("Virtual Reality") OR TI=("Virtual Reality")) AND (TI=("Information Retrieval") OR TI=(search) OR TS=("Information Retrieval") OR TS=(search)) AND (TI=(system) OR TS=(system)) NOT TI=(Augmented AND Reality) Databases=WOS, KJD, MEDLINE, RSCI, SCIELO Timespan=2013-2020*

- Query for *Scopus*: *TITLE-ABS-KEY (information retrieval OR search) AND TITLE-ABS-KEY (virtual reality) AND TITLE-ABS-KEY (system) AND PUBYEAR > 2012 AND PUBYEAR < 2021 AND NOT TITLE augmented reality*

As there is a hard limit of 1000 results for any query in *Google Scholar*, multiple queries limited to each year (2013 to 2020) were applied. Also, the query for *Google Scholar* is more strict, as it combines all terms with the Boolean *AND*. Only so was it possible to get less than 1000 results for each year. The query for *Web of Science* uses the above-mentioned databases and limits results to a combination of the terms Virtual Reality, system and Information Retrieval in either the publication title or as its topic while excluding research on Augmented Reality. The query for *Scopus* adopts the same strategy for title, abstract or keywords. These differences in the queries are only due to the heterogeneous design and structure of the three databases.

After identifying all relevant works of research based on these queries, the full-texts of these works were analysed for additional terms that ought to be included in the queries. And while many works included more specific terms such as *immersive search*, *search engine* or *3D object retrieval*, no additional common terms could be identified. Finally, all works cited in relevant results and the other works from relevant authors were also analysed and yielded no additionally keywords as well.

## 2.5 Selection of sources of evidence

To make the selection of papers relevant to the research questions, several of criteria must be met. These can be summarised as a methodical screen as suggested by [13]. The methodical screen sets the following requirements:

1. The work deals with something designed especially for VR

2. The work does not deal with something designed exclusively for AR

3. The work deals with something containing search functionality in VR

4. Used hardware: *Oculus Rift* DK 1 or newer

5. Publication type: paper, poster paper or thesis with or without study

To apply the methodical screen, the title and abstract of the resulting papers were read. If the entry was entirely unavailable because it was retracted, not written in English or not scientific work, it was excluded and this was noted in the item *valid*. If there was no abstract available after the automated search, but a DOI or unique title, the abstract was retrieved manually as part of the full text.

Only papers which passed the first two points (designed for VR, not designed for AR) were further considered. Systems can be theorised or software prototypes. Next, papers needed to pass the next point (have/discuss search functionality in VR). Specifically, the system must feature or discuss systems or system concepts, which use some user input to retrieve a set of relevant documents and displays them. The entire interaction between system and user must happen in Virtual Reality. Augmented Reality and Mixed Reality systems are therefore excluded. All papers were reviewed by the author of this paper. If the relevance of a paper was not entirely certain, it was still included for further analysis to not exclude potential relevant results. From here on, papers were retrieved in full and skim read to ensure the first three points of the methodical screen were assessed correctly and to assess the remaining points (paper type and used hardware).

**2.5.1 Browsing vs. searching a system.** An important distinction between the actions of browsing and searching in a system must be made. As [27] puts it: "[. . .] we distinguish between two fundamental retrieval approaches: searching or browsing. A user either systematically searches for documents, or he browses through document collections". So, browsing is a user interaction based on predefined (e.g. random or personalised) collections. Examples are personalised Social Media feeds or digital exhibitions. The user navigates the collection by scrolling or with pagination and selects relevant elements for a detailed view. A user searches a collection if they actively input their query or define filters. The third point of the methodical limits results only to systems with user search functionality.

## 2.6 Data charting process

For *Web of Science* and *Scopus*, the aforementioned queries were entered via the web interface and downloaded directly with the export function of these systems. For *Google Scholar*, a tool

was used to extract all results to the given query programmatically. The data was cleaned up, adjusted to the same format, encoding and limited to relevant attributes (author, title, DOI, publication year and abstract) and exported into spreadsheet software for relevance assessment (practical screen). All data was then reviewed by one reviewer, who added the first assessment of relevance by determining if the work focused on Virtual Reality and a search system at all. This was deferred from the title and the abstract. Ambiguous cases were kept for a more detailed assessment. The final assessment included the retrieval of the full paper by the same reviewer. All assessments were recorded as data attributes with the spreadsheet software.

### 2.7 Data items

Besides the basic information about the paper (author, title, DOI, publication year and abstract if available) the following items were gathered:

- vr_a
- search_a
- valid
- vr_b
- search_b
- explanation

The first two items aim to infer from the paper title and abstract whether the work is about VR at all (*vr_a*) and whether it focuses on a search interface (*search_a*). If there was any reason to exclude the work based on the practical screen, this was noted in the item *valid*. After the full work was taken into account, the items *vr_b* and *search_b* reassess whether the work really focuses on a Virtual Reality system and whether it has search capabilities respectively. Finally, the item *explanation* gives a reason if the system did not meet these requirements after all.

### 2.8 Critical appraisal of individual sources of evidence

As this review aims to survey the body of research relevant to the research questions, a critical appraisal of the overall quality of the works is not necessary. However, the scope and quality of the evaluation of prototyped or theorised systems varies greatly and is important for RQ5. Therefore, the evaluation methodology is broadly assessed and discussed.

### 2.9 Synthesis of results

Simple tables were used as data extraction forms as suggested by [13]. Regarding to the research questions these tables answer the following questions:

1. What type of research was conducted?

2. What specific kind of VR hardware was used?

3. What kind of user input technology was used?

4. What is the main concept for user interaction?

5. What retrieval features are available in the system?

6. How are the search results represented?

7. Is there an evaluation and if so, how was it employed?

8. If applicable, what are the main evaluation results?

For the first item, [13] suggests differing between qualitative and quantitative research. For the given rationale, it makes more sense to focus on the general topic and type of the paper. Papers are categorised as follows:

- brainstorming specific systems in theory

- brainstorming the topic in theory

- report on a working prototype

- provide an overview of systems on the topic

For item five, common retrieval features were collected from [28–30].

Research questions 2, 3 and 4 are closely connected to User Experience Design, therefore it is of great interest to this review how user input and result presentation is actualised in the examined systems. For all papers with screenshots, copies of the interfaces screenshots are included.

Also, especially interesting and important concepts are summarised and discussed in the text. Finally, based on the summary of the examined systems a comparison to traditional Information Retrieval systems is made (RQ5).

## 3 Results

### 3.1 Selection of sources of evidence

Fig 3 shows the absolute numbers of reviewed documents and the remaining documents after a total of three selection stages.

To some extent, the practical screen could be applied automatically via the database queries to ensure the correct publication date and the occurrence of keywords. This resulted in 1042 documents being potentially relevant to the research questions after excluding duplicates.

The first stage affirms the practical screen and applies the methodical screen intellectually by looking at title and abstract only. This enabled a swift reduction of the result set from 1042 to 123 documents. The second stage retrieves the full text of each document and ensures that Virtual Reality is the primary or a major topic. The same goes for Information Retrieval as a central topic in general and search interfaces specifically. The full data table S1 File [31] gives a reason for each document excluded at the second stage (*explanation* column). This resulted in 30 relevant documents grouped into 16 distinct research efforts shown in Table 1. A grouping was only applied for identical systems/concepts by the same authors or an intersection of authors.

15 relevant documents were found via *Scopus* and 19 documents via *Google Scholar* (including an overlap of nine). Only one of the final documents was found via *Web of Science* and even that document was found via *Scopus* as well. Eight articles were identified from the references of already included articles or other works of authors of already included works.

### 3.2 Characteristics of sources of evidence

Table 1 summarises all relevant works concerning to the research questions. Works about the same or updated system by the same author(s) are grouped. For these, the *research type* broadly classifies what kind of research was done. For this review, a paper is a system paper if it focuses mainly on the description and presentation of a conceptual or prototyped system. A theory paper discusses relevant concepts in theory or theorises a possible system. A general review does no original work on its own but collects and summarises existing work on the topic. Next, the technologies used are summarised in the table as well. *VR hardware used* is self-

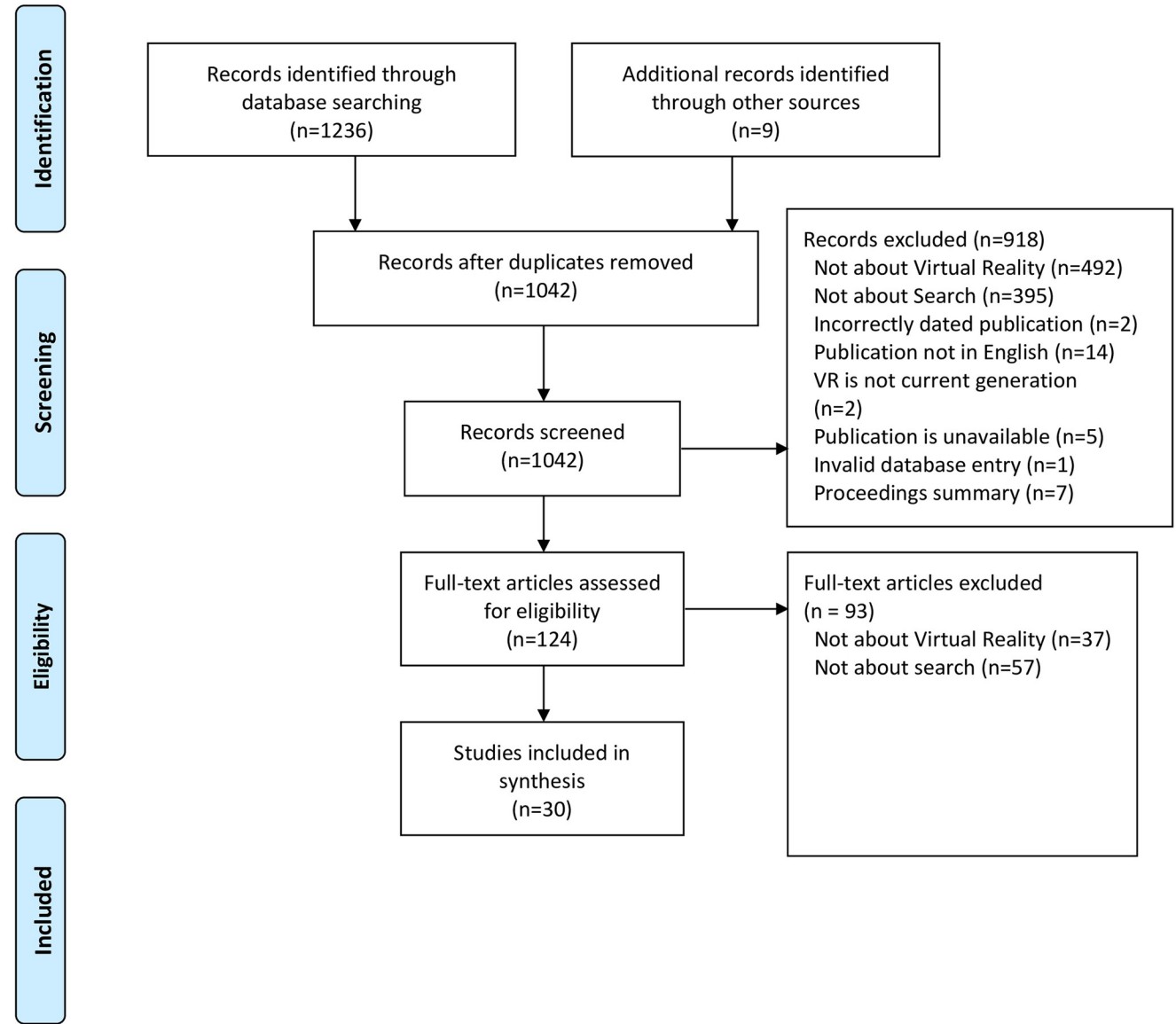

**Fig 3. Flow diagram of review steps following the PRISMA guidelines from [10].**

explanatory and includes the entire systems, and thereby the head-mounted-display and tracking technology (e.g. the *HTC Vive* system). *User Input technology* is listed separately as often additional hardware is used that is not part of the commercial system (e.g. the *Leap Motion* hardware). The *User interaction concept* briefly and broadly summarises the general functionality of the system regarding user input and interaction (e.g. using grabbing gestures to select items). Finally, the *result representation* briefly summarises the general concept to display search results (e.g. 3D objects in a cylindrical grid).

### 3.3 Results of individual sources of evidence

As a key result of this review, the answers to RQ1, RQ2 and RQ4 are also summarised in Table 1. There are 16 distinct systems or theories detailed in a total of 30 research items. These

**Table 1. Summarised data on all relevant works grouped by distinct systems or theories.**

| System/ Theory | Research type | VR hardware used | Main use | User Input technology | User interaction concept | Result representation |
|---|---|---|---|---|---|---|
| A ([32, 33]) | System paper | Oculus Rift | Retrieval of academic literature | Voice input + Hand Tracking (Leap Motion) | Gestures, select, Query by voice | 3D objects |
| B ([34]) | Review | Oculus Rift | 3D content retrieval serendipity effects | Various | Not applicable | Not applicable |
| C ([35–42]) | System paper | OG HTC Vive | Retrieval of Lifelog Images | HTC Vive Wand controllers | Select, filter, typing | 2D Image grid in 3D space |
| D ([43]) | Theory paper | Not applicable | 3D object retrieval | None | Gestures | 3D shapes, but not focus |
| E ([44–46]) | System paper | Okulus Rift CV 1/DK1 | 3D object retrieval | Oculus Touch controllers/Leap Motion | Sketch | 2D grid of 2D thumbnails of objects. Full 3D render of selection on the floor |
| F ([47, 48]) | System paper | Okulus Rift DK 1 | 3D object retrieval | Voice recognition + Leap Motion/SPF device | Query by voice, gestures | 3D objects in different grid: rect., square, cylinder, sphere |
| G ([49]) | Theory paper (system proposal) | Not applicable | Book retrieval (library) | None | Virtual OPAC + interactive visualisations | Virtual Books |
| H ([50, 51]) | System paper / thesis | OG HTC Vive | Retrieval of Lifelog Images | HTC Vive Wand controllers | Fly, teleport, select | Display 2D images on a world map (2D or 3D) based on GPS |
| I ([52]) | System paper | Google Cardboard | Image Retrieval | Voice recognition, gaze | Voice commands, select | 2D images, tags and a map in a curved grid |
| J ([53]) | System paper | Oculus Rift S | Multi-user 3D object retrieval | Voice recognition, Leap Motion | Voice commands, query with gestures | 3D objects representing database elements |
| K ([54]) | System paper (not yet published) | Okulus Rift DK 1 | 3D object retrieval | Wiimote Controller | Virtual paint brush and palette, select | 3D objects on a painting palette and a 3D model in space |
| L ([55]) | Theory paper | Not applicable | Possibilities of VR for IR systems | Hand tracking, controllers, (virtual) keyboards | Query input, query by sketch, query by example | Information visualisations |
| M ([56]) | System paper | OG HTC Vive | Image Retrieval | HTC Vive Wand controllers | walk, teleport, select, filter | Groups of 2D images in circular grids |
| N ([57]) | System paper | OG HTC Vive | Retrieval of software repositories | HTC Vive Wand controllers | walk, select, grab | 3D Planets representing software repositories colour, position and size encode information |
| O ([58, 59]) | System paper | Unknown, but current gen. | Retrieval of text (natural language) | Leap Motion | Gestures, typing | Curved floating 2D text |
| P ([60, 61]) | System paper | Oculus Quest | Analyse effect of spatial position of search results | Oculus Quest controllers | Select | Text in 2D list, 3D grid or 3D arc |

are labelled as systems A to P. A total of 17 of the analysed works feature some form of evaluation of prototyped systems (not grouped by systems). All entries feature either functioning systems (n = 12), a review (n = 1) or theoretical concepts (n = 3) of search systems in Virtual Reality with current generation hardware in mind. Additionally, we can also see a trend towards the popular commercially available hardware, which is intended for end-users. Seven systems use one of the *Oculus (Rift)* headsets, four use the *HTC Vive*, one system uses a solution based on *Google Cardboard*. For one system the hardware could not be determined but is heavily implied to be current-generation in the text (system O, see [58] and [59]).

RQ3 asks which specific Information Retrieval features are utilised in Virtual Reality Information Retrieval systems. For the analysed systems Table 2 lists which Information Retrieval

**Table 2. Overview of specific Retrieval functions present in the analysed systems.**

| System | Boolean Operators | Phrase Searching | Match Exact Words/ Phrases | Field Specific Searches | Limit Field Searches | Proximity Search | Range Searching | Thesaurus Or Permuted Index | Subject Search | Query By Example | Filtering | Relevance Ranking | Fuzzy Match | Browsing |
|---|---|---|---|---|---|---|---|---|---|---|---|---|---|---|
| A | | x | x | | | | | | | | | | | x |
| C | | | x | x | x | | x | x | x | | x | x | | x |
| E | | | | | | x | | | | x | | x | x | x |
| F | | x | x | x | | x | | x | x | | x | x | | x |
| H | | | | | | | | | | | | | | x |
| I | | | | | x | | | x | x | | | | | x |
| J | | | | | | x | | | | | | | | x |
| K | | | | | | | | | | x | | | x | x |
| M | | | | | | | | x | x | | x | x | | x |
| N | | | | x | | | | | | | x | | | x |
| O | x | | x | | | | | | | | | x | | x |
| P | | | | | | | | | | | | | | x |

features are supported by which systems. Meanwhile, some retrieval features that were identified as common by [28] were not supported by any of the analysed systems:

- Save Search

- Search History

- Truncation

- Wildcard

- Have Rules Of Precedence With Nested Queries

- Stemming

Table 3 summarises the evaluations and their most important results. Two systems are without any evaluation (system J [53] and system N [57]). For the others, the number of test subjects varies overall between five and 36 people. 12 works provide demographic information. Here, the overall age ranges between 17 and 65 years. If gender distribution is given, there is an imbalance towards male participants. Many evaluations were conducted with novice VR users. 12 of the 16 works conducted task-based evaluations with these users. Two additionally applied interviews and profiling. Three other works instead evaluated by partaking in an Information Retrieval challenge with other systems, by letting users rate their impression on a 5-point Likert scale or by using established quantitative measurements for IR systems (e.g. measuring the number or rate of relevant results). Ten of the works feature evaluations that directly compare user task efficiency, effectivity and satisfaction with some kind of baseline system, typically a conventional desktop search system controlled with mouse and keyboard.

The theoretical works, while more abstract by nature, provide ideas and concepts relevant to Interactive Information Retrieval Systems in Virtual Reality as well. First, [34] pictures the possibilities of exploratory search and serendipity effects in such systems. As such effects are typically present with physical mediums and locations (especially libraries) they see them threatened by digitalisation. VR search systems could potentially compensate for this effect. Exemplary mentioned are VR workstations in libraries and access to 3D models based on real data (e.g. from Architecture, Anthropology or Bio-Chemistry). [43] elaborates on this, as they see 3D object retrieval as an especially promising application of VR technology. They ask for more work to be done, especially by using natural gestures for interaction and by providing adequate data sets as well. Again regarding physical libraries, [49] elaborate on the potential design on a Virtual Reality library retrieval system, which should feature a virtual library, data visualisation for result representation and thereby "enables readers to participate in the whole process of information retrieval and browsing". Lastly, [57] theorise on all aspects of such systems, including user input and interaction and data visualisation. They call for more research in the area especially considering the ever-growing number of Internet of Things devices and the data they produce.

## 3.4 Synthesis of results

Based on the assumption that the review was able to extensively survey the relevant body of research, all items in Table 1 are relevant to answer RQ1 (What prototypes and concepts of Virtual Reality search systems with current generation hardware exist?). In short, systems (concepts) A to P are, based on the methodology of this review, the only Virtual Reality search systems (concepts) with current generation hardware.

**Table 3. Summarised data of all works featuring evaluations.**

| Research item | No. Of users | Demography | Evaluation type | System(s) for comparison | User task efficency (speed) | User task effectivity (quality) | User system satisfaction/ preference |
|---|---|---|---|---|---|---|---|
| [32] | 5 | Unknown | Task solving | Non-VR same system and same data (web) | VR | Non-VR | VR |
| [35] | 12 | Unknown | Task solving | 2 VR, 1 non-VR system (web) | equal | Not measured | VR (1 mode) |
| [36] | - | - | - | - | - | - | - |
| [41] | 16 | Various technical backgrounds | Task solving | Non-VR and VR | equal | equal | VR |
| [37] | 16 | Solid in English, rudimentary computer skills | Task solving, Comparison of VR | Multiple variations of VR interfaces | Not applic. | Not applic. | Not applic. |
| [38] | - | - | - | - | - | - | - |
| [39] | 16 | Computer users, limited VR experience, | Task solving | Conventional 2D system | Non-VR | Not measured | VR |
| [42] | - | - | - | - | - | - | - |
| [44] | 30 | 20m 10f, avg. age 26, members of research department and general public | Task solving | Other interaction method (VR), browsing entire catalogue | Not applic. | Not applic. | Not applic. |
| [46] | - | - | - | - | - | - | - |
| [45] | 5 | 4m 1f, 25-43 age, with VR experience | Task solving | Sketching on a real tablet | VR | VR | VR |
| [40] | 6 | Novice and expert groups | Lifelog challenge | Not applic. | Not applic. | Not applic. | Not applic. |
| [48] | 29 | 17-40 age, 90% CS students | Task solving | Some basic 2D system | equal | Not measured | VR |
| [47] | 20 | 18-50 age, 16m 4f | | Commercial non-VR system | VR | VR | VR |
| [50] | - | - | - | - | - | - | - |
| [52] | 11 | Age group of 21-30 | Scoring on 5-point-scale | None | Not applic. | Not applic. | Not applic. |
| [53] | - | - | - | - | - | - | - |
| [33] | 10 | Unknown | Task solving | Non-VR variation of same system with same data (web) | VR | Non-VR | VR |
| [54] | 20 | 3f, 17m, age 22-65 | Task solving, Profiling | Non-VR 2D retrieval system for 3D objects | VR | VR | VR |
| [51] | 35 | 16 already used system before, 2/3 m 1/3 f, age 21-33 | Task solving, interviews | Multiple variations of VR interfaces | Not applic. | Not applic. | Not applic. |
| [56] | 20 | Age 18-20 years | Task solving | Non-VR system | Non-VR | Non-VR | VR |
| [57] | - | - | - | - | - | - | - |
| [58] | ? | Unknown | Recall, Preci- sion@10, MAP, nDCG@10 and a-nDCG | Not specified | Not specified | Not specified | Not specified |
| [59] | - | - | - | - | - | - | - |
| [60] | - | - | - | - | - | - | - |
| [61] | 36 | 15m, 22f, students and university employees | Task solving | Other result presentations (list, grid, arc) | Not applic. | Not applic. | Not applic. |

As for RQ2 (How are user interaction and especially user input realised in these systems?), the answer is twofold. On a technological side, we see seven systems that are using controllers with positional tracking and buttons that are included with the VR system. Five use the Leap Motion system to enable hand tracking, while four systems allow voice input. Other input methods such as gazing (looking at elements for certain amounts of time) and more specialised hardware or the Wiimote controller are used by one system each. These numbers include an

overlap as sometimes multiple input methods were used by the same system. On the other hand, the (primary) design concepts to handle the user interaction based on these technologies are as follows (again, including overlap): At least four systems use controllers for hand gestures or selections. Three systems utilise speech recognition, two systems let users sketch in 3D space with controllers. For movement, two systems allow teleportation, while one system uses physical walking with full positional tracking. Finally, zooming, grabbing and filtering are utilised at least once by each system. This list is not entirely complete, as some works gave little insight to their interaction concepts and as there are several software frameworks available for the used commercial VR hardware that may or may not already include such features. For the three theoretical works, user interaction is abstractly discussed. This includes gestures in general, virtual catalogues (e.g. library OPAC), interactive visualisations and direct query input or query-by-example.

The more abstract RQ 3 (What Retrieval features are used in systems?) can be summarised from Table 2. We can see that all systems feature some kind of browsing experience, but for systems J and P, browsing is the only of the common retrieval features they support. Only system O supported the use of boolean operators like *AND/OR* to combine keywords in queries. Only system C allowed to limit an attribute to a certain range of values. *Relevance ranking of search results* was the second most used Retrieval feature (n = 5), followed by the matching of exact words or phrases, the use of a thesaurus or permuted index for searching, subject search and filtering (n = 4). The other Retrieval features were less common (n = 3 and n = 2). Some common retrieval features were not present in any of the analysed systems (6 features as listed above).

Regarding RQ4 (How are search results displayed in these systems?), five systems represent search results as various kinds of 3D objects (systems A, E, F, J and K). Four systems use 2D images (systems C, H, I and M) and three systems (systems G, O and P) retrieve some kind of text (entire books and natural texts). Finally, one system (system N) retrieves entire software repositories. The theoretical works discuss 3D objects (systems B and D), as well as graphs (system L).

Fig 4 illustrates the different approaches to user interaction and result representation design used by the various system prototypes as available. All sketches are based on screenshots provided in the analysed works and make comparison easier (as the camera angle and image quality of the screenshots varied). Only system P didn't include a screenshot. Instead, a sketch of the broad concept for the user interface is included.

On an abstract level, we can also group these systems by the similarity of their approach in user interaction and result presentation (RQ2 and RQ4). Systems A, J and O both feature hand tracking and hand gestures. But, in system O there are only some innovations compared to a classic Information Retrieval system. It uses a virtual keyboard and gestures to select text and apply boolean query operators. The system sticks to known conventions for non-VR systems by displaying 2D text on a virtual curved 2D screen. Systems A, F and N on the other hand allow direct interaction with 3D objects. For system A, these objects are spheres in a network graph. System F displays virtual *Lego* blocks and also encodes relevance by colour. System N also allows interaction with objects, but it utilises a planet metaphor. Both, system E and K let users sketch objects with a virtual paintbrush. They use these 3D sketches to retrieve the closest matching 3D object in catalogues with an algorithm. Both also let users select stroke colour and provide additional information on a virtual painting palette. Systems C, I, H and M focus on Image Retrieval, systems I and C let users select relevant tags, system H utilises the GPS information to display the images on a world map and finally, system M groups similar images and places them in circles around the user.

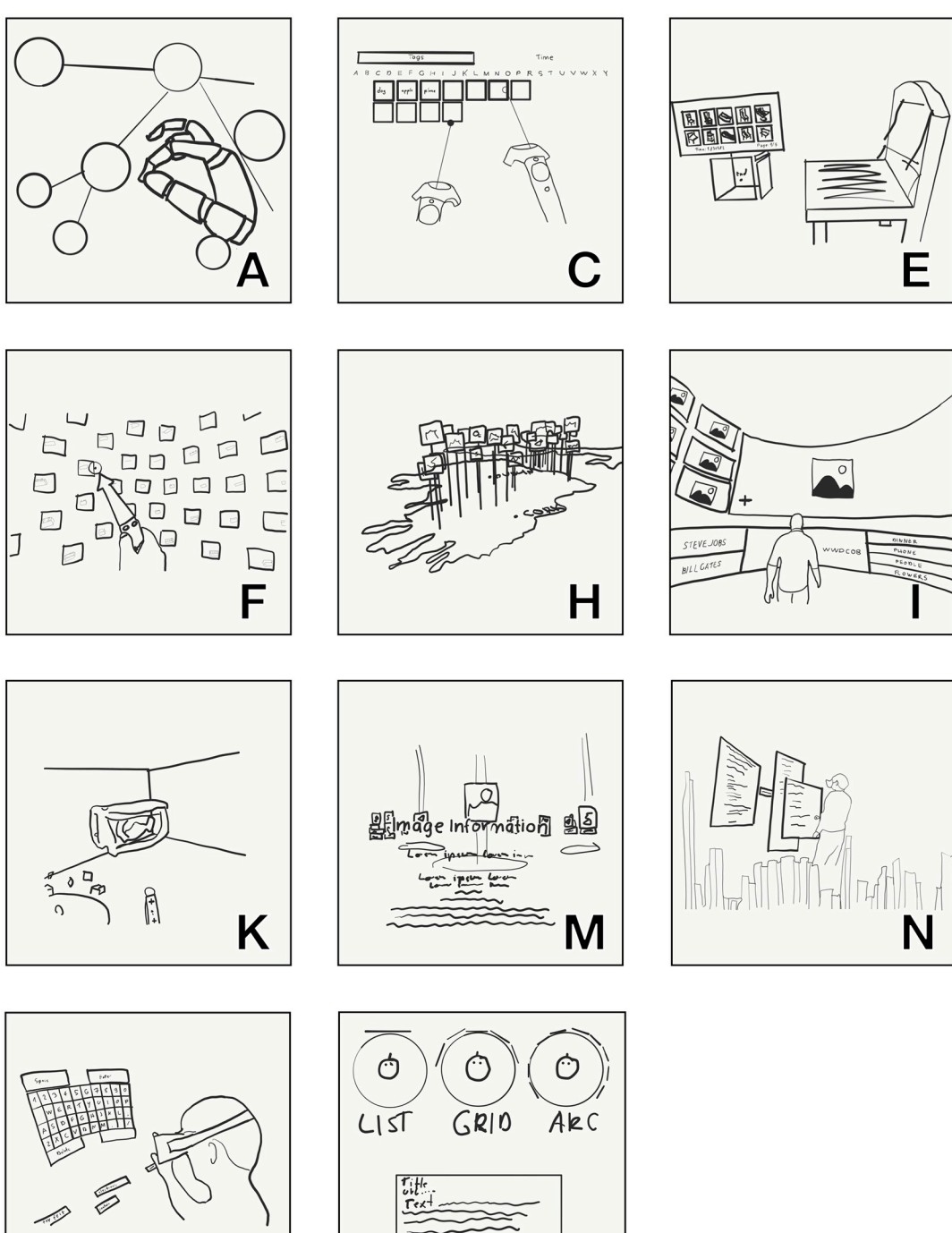

**Fig 4. User interface design approaches of various system papers.** Letters correspond to Table 1.

RQ 5 asks whether these systems can compete with conventional (2D) search systems or not. Out of the 10 papers that did comparisons, five reported a significantly better user efficiency (speed) for their VR system compared to either a more basic VR system or a conventional 2D retrieval system. Three systems were equal to such systems and two performed worse. Seven papers also evaluated user retrieval effectivity (quality) against the aforementioned baseline systems. Three reported superior task effectivity with their VR systems (three reported worse, one equal). All 10 papers with comparative evaluations report significantly higher user satisfaction or preference for their VR systems.

However, there are additional challenges of using Information Retrieval Systems in Virtual Reality (in comparison to conventional systems). Some of these challenges are also acknowledged by the reviewed works. First, a user has to value the benefit of the system high enough to merit starting or setting up the Virtual Reality Environment they supposedly have at their disposal. While there has been a decrease in cost and recent newfound popularity of Virtual Reality [2], as of today, Virtual Reality hardware is not yet a commodity that can be found in nearly every household or workplace [62]. The setup of such systems can range from simply powering up and putting on a stand-alone Virtual Reality headset (with or without controllers) to connecting a headset to a computer, powering the computer, headset and additional tracking hardware as well as controllers and starting a special software on the computer. In any case, it is more time-consuming than picking up a smartphone or using a computer alone. For some, prolonged usage of possibly heavy VR googles may cause physical discomfort or pain [51, 54]. Depending on the hardware and the design of the virtual environment, the user can just sit at their desk or would need enough unoccupied physical space to walk multiple steps in either direction. [51] (system H) discusses these limitations in detail but also envision many of these limitations to be resolved by improved hardware as commercial Virtual Reality hardware currently develops to be easier to use [5]. Other limitations are more native to the medium itself. These include the strong immersion in the virtual environment and the resulting disconnect of the user from their physical space (in contrast to Augmented Reality) which limits the users' mobility and the usage in public spaces. A possible conclusion was suggested by one of the users of [56] system: "For short term tasks, non-VR will be better as it allows someone to sit down and start using it but for long term tasks, VR has a significant advantage because of the features it provided." Other scenarios would have the user already immersed in a Virtual Reality Environment, using software for productive or recreational purposes. The user now may want to use an IR system to search for certain items. The limitation lays in the current design of underlaying software of commercial Virtual Reality systems. The established metaphors for multi-tasking and context-switching in graphical operating systems and web browsers (desktops, applications, windows and tabs) are not fully established (or replaced) in Virtual Reality environments [34]. In some cases, the user can only fully use one Virtual Reality application at a time and needs to start and close applications one after the other (for example in SteamVR environments). In others, multi-tasking is only possible with the established conventional 2D metaphors on a virtual screen that overlays the Virtual Environment (for example in Oculus environments). [58] (system O) state, that "[m]ost users might not be ready to use VR. [. . .] sometimes they feel exciting, other times they feel challenged and confused." But they also report on measures to help users with none or little experience in virtual environments, including animations and highlighting that illustrate the correlations between user input and the consequences. There is also evidence for positive psychological effects of the virtual environment that can not be observed with traditional displays and systems [63]. As [53] point out, Virtual Reality could provide a natural user interface and allow users to explore and interact data in a more natural way (compared to tabular data presentation). And current adoption rates of commercial Virtual Reality hardware and the overall user experience of such systems

are not primarily academic problems and don't impede on the promising results of the evaluations conducted by some of the reviewed works. In matters of design and usability, researchers have already started to point out open research issues and ask for participation [43, 55].

Finally, traditional Information Retrieval systems are limited as well. The option for the user input the options are more limited compared to Virtual Reality systems [33] which can include tracked controllers, hand tracking or virtual keyboards. Conventional flat displays have limited screen space compared to the theoretically infinite virtual space in VR [32]. 2D displays can also not provide adequate depth perception for 3D content. Many of the analysed works include 3D objects in their results (see Table 2). [54] elaborate on how traditionally 3D objects are retrieved by searching in the metadata which is limited and prone to errors. By querying with an abstract sketch that is created by the user in a short time in Virtual Reality, results can be much more relevant to that user. Generally, traditional Information Retrieval systems can have overloaded interfaces and limited user interaction [32]. [34] suggest that serendipitous Information Retrieval, the unintended discovery of relevant information, can more easily occur in Virtual Reality experiences that mimic information retrieval in the real world (for example in libraries).

### 3.5 Critical appraisal within individual sources of evidence

A full appraisal of all analysed works, possibly with a scoring system, lies beyond the scope of this review and is not needed to answer RQ1 to RQ4. However, to answer RQ5 the evaluations conducted in the analysed works should be valid and sound. Therefore, these evaluations were broadly appraised. Here, the methodology varies greatly for the 18 studies that conducted some form of evaluation. First of all, for the 11 studies that provided demographic information, only six disclose the gender distribution of participants. In all but one of them (system P [61]), the distribution was skewed towards male participants (66%, 80%, 80%, 85% and 66% respectively). Overall, based on the median or mean (as available) of the age of participants of the seven studies that disclosed this data gives broadly a mean age of 27 years (although ranging from 18 to 65 years). On another topic, [64] elaborate in great detail, that the evaluation of complex visual systems has many design implications and can be challenging. They state that the outcome of such evaluations can be based on foregone conclusions, that experiment setups can be faulty in the first place and that results can be misinterpreted or exaggerated. 10 of the 17 studies that are analysed in this paper, conducted evaluations with a comparison to a baseline system, like conventional search systems or a more basic VR system. Overall, for user task efficiency and effectivity eight out of 17 times the VR systems of the authors performed better or was rated better. Additionally, all systems were evaluated to have better user satisfaction or preference. While potentially a promising prospect for the potential of successful Information Retrieval Systems in Virtual Reality, this could also hint at what [65] calls "using a very poor alternative method as a control, thereby exaggerating the value of their own method". When this is combined with a small number of test subjects, the results might not be applicable for a wider group of people.

## 4 Discussion

### 4.1 Summary of evidence

The databases *Google Scholar*, *Scopus* and *Web of Science* were queried between January and February 2020. After applying practical and methodical screens, a total of 1042 unique works resulted in 30 relevant documents for the given research questions. These could be reduced to 16 distinct systems and concepts. These systems use commercially available hardware like the *HTC Vive*, Google Cardboard or *Oculus* device (including different device models) (RQ1). The

works discuss and utilise several user input technologies, ranging from voice input to hand tracking or the use of controllers. These technologies allow users to use gestures, draw sketches, query with speech, teleport themselves, select or type to interact with the environment or retrieve items (RQ2). Although conventional retrieval features are less commonly used (except for browsing and relevance ranking), systems enable retrieval of academic literature, 3D objects, lifelog data, images, books and texts (RQ3). These items are displayed as 3D objects, 2D images, planets, text virtual books and arranged in grids, arcs, maps, cluster, spheres or (curved) rectangles (RQ4). Finally, many of these systems were compared to conventional counterparts, and in these evaluations, user task times were shorter or equal (eight out of 10), they solved tasks with good performance (four out of seven). Users always were more satisfied with the Virtual Reality systems (10 out of 10) (RQ5).

## 4.2 Limitations

A general limitation when carrying out reviews on interactive software systems is the limited presentation of these interactions in a research paper. Coloured images (screenshots) are still the default means of providing the reader with either an overview of a system as a whole or to illustrate an important concept. It is not common to include a reference to a video which showcases these systems, although this becomes more popular. A reference to any kind of working software code is an exotic rarity. Therefore, it can be a challenge to correctly analyse the quality of a systems regarding the user experience. Of course, research in the field should not be based on the subjective impression of the reader, but some kind of quantitative evaluation, most commonly based on user feedback. Still, an expert could understand things like a plausible correlation between low user performance or satisfaction and poor implementation of readable text by examining the software code or a video (one video was provided by [57]).

Regarding this review, 6603 of the potentially less relevant *Google Scholar* results were not assessed intellectually. However, this was compensated by reapplying parts of the practical screen to this result set even more strictly and by additionally including most "relevant" items based on *Google Scholars* internal ranking for each year.

All three queries send to the literature databases, require works to include the term *system* in either title, abstract or topic (or anywhere in the text due to the limitation of the Google Scholar query language). This term is commonly used in the context of software development and even theorised design and therefore well-chosen regarding the research questions. However, while not indicated by exploratory queries conducted before this review, there might still be some works that don't use the term at all but are relevant nonetheless.

Some works ([50] and [51]) feature a system that fits the requirement of an Information Retrieval system as they use an abstract query, the user's interaction with a world map, to represent results (images geotagged to the location). However, the system interaction is described as browsing in these works (but the authors also included the term *search* prominently). Possibly, some other works would be of interest regarding the research questions but were talking exclusively about *browsing* and were therefore blocked by the practical screen already. Nonetheless, this review successfully identified the relevant body of research regarding the research questions where authors intended the system to provide search functionality and result representation in an interactive Virtual Reality environment.

## 4.3 Conclusions

Regarding the information sources of this review, there are of course other sources of insight beyond purely scientific works. Future work could include a general survey of sources such as Social Media, online news, blogs and so on. Additionally, the biggest online stores for

consumer VR software could also be searched (including [22] and [24]). As this is a time-consuming task, it seems sensible to make this effort based on more narrow research questions or while designing a working prototype as mentioned before.

This review was able to comprehensively probe the body of scientific literature to identify 30 research works. These were summarised as 16 individual systems or theories about Interactive Information Retrieval systems in Virtual Reality following the announcement and release of the *Oculus Rift* and other current generation hardware systems. There were different concepts for user interaction, ranging from gesture-based interaction with objects, like combining keywords or selecting results, to traversing a world map or virtually sketching approximations of desired objects. Results were visualised as curved text, flat, spherical, curved or cylindrical image grids, planets or floating 3D objects. The majority of comparative evaluations of the systems found significantly better user performance in either speed or quality or at least a higher user satisfaction or preference.

Although promising, many of these developments are the very first iterations of their kind. Therefore, future work should also focus on the design of working prototypes based on the findings of this review. Such systems should feature proven user input technology, user interaction paradigms and result representation. For Information Visualisations for example, [65] describes a "visual grammar" for the elements of node-link diagrams and geographical maps. Regarding graph visualisations, it is generally accepted that nodes of some kind that are visually connected also share some kind of relationship conceptually. Could a similar grammar be created for the elements of a Virtual Reality search interface?

On another note, as [43] correctly assess, there is a general lack of ready-to-use and well-fitting data sets which can show the strengths of the medium Virtual Reality. Future work could also identify and even modify existing data sets for use in Virtual Reality software environments. Finally, this review also identified that rigorous and open-minded evaluation with an acceptable baseline for comparison and equal gender distribution of study participants is important.

## Supporting information

**S1 File. CSV file with all analysed papers.** The smaller number of items is due to the automated removal of some duplicates before intellectual analysis.
(CSV)

**S2 File. PRISMA ScR checklist.**
(PDF)

## Author Contributions

**Conceptualization:** Maurice Schleußinger.

**Data curation:** Maurice Schleußinger.

**Formal analysis:** Maurice Schleußinger.

**Investigation:** Maurice Schleußinger.

**Methodology:** Maurice Schleußinger.

**Resources:** Maurice Schleußinger.

**Software:** Maurice Schleußinger.

**Validation:** Maurice Schleußinger.

**Visualization:** Maurice Schleußinger.

**Writing – original draft:** Maurice Schleußinger.

**Writing – review & editing:** Maurice Schleußinger.

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
