## [Decision Letter · Decision Letter 0]

24 Nov 2020

PONE-D-20-32580

Information Retrieval Interfaces in Virtual Reality - A Scoping Review Focused on Current Generation Technology

PLOS ONE

Dear Dr. Schleuβinger,

Thank you for submitting your manuscript to PLOS ONE. After careful consideration, we feel that it has merit but does not fully meet PLOS ONE’s publication criteria as it currently stands. Therefore, we invite you to submit a revised version of the manuscript that addresses the points raised during the review process.

We look forward to receiving your revised manuscript.

Kind regards,

M. Usman Ashraf, Ph.D

Academic Editor

PLOS ONE

Journal Requirements:

2. In your study selection flowchart (fig. 3), please specify the reasons for exclusion at each step of selection.

3.Thank you for stating the following in the Funding Section of your manuscript:

[I acknowledge support by the Heinrich Heine University Duesseldorf.]

 [The author received no specific funding for this work.]

4. Your abstract cannot contain citations. Please only include citations in the body text of the manuscript, and ensure that they remain in ascending numerical order on first mention.

Reviewers' comments:

Reviewer's Responses to Questions

**Comments to the Author**

1. Is the manuscript technically sound, and do the data support the conclusions?

Reviewer #1: Yes

Reviewer #2: Yes

2. Has the statistical analysis been performed appropriately and rigorously? 

Reviewer #1: Yes

Reviewer #2: Yes

3. Have the authors made all data underlying the findings in their manuscript fully available?

Reviewer #1: No

Reviewer #2: Yes

4. Is the manuscript presented in an intelligible fashion and written in standard English?

Reviewer #1: Yes

Reviewer #2: Yes

5. Review Comments to the Author

Reviewer #1: The author provided a solid study on interactive information retrieval systam in Virtual Reality following the announcement and release of the current generation hardware system. It pre-designed four questions, covering the prototype/concepts, user interaction, retrieval features, and synthesis/display. After analysis a mount of related works, it present solid study results in corresponding to those questions. The data itself is solid and comprehensive. Moreover, it provided all the annotated data in the attachment. Consequently, I prefer to accept it for publication.

Reviewer #2: This article is an analysis of the state of the art related to Information Retrieval Interfaces in Virtual Reality. The manuscript is well organized and comprehensively described and with adequate references. In a high level, the paper tries to answer to five questions. The reviewer appreciates the effort and time spent for collecting and reviewing literature on the use of VR IR systems. However, the reviewer thinks the paper should be improved:

- In the abstract mentioning just the Google Scholar, Scopus and Web of Science could be problematic because it is not clear why IEEE and ACM where not included.

- Related to the keywords, other terms could be incorporated, like “Head Mounted Display”

- The introduction needs to focus on the topic and goals/contribution of present research. I think that explaining the concept and history of “Virtual Reality” is too much. This is not a survey of general VR technology.

- In Methods: More of information about PRISMA should be included. Please specify what it represents and how it benefits this study. What is the reason for including student theses?

- In the Synthesis of results section review also the challenges and their implications in using VR IR systems compared to traditional (non-VR) IR systems.

- The English language and style is good and clear to be understood.

Regards

6. PLOS authors have the option to publish the peer review history of their article (what does this mean?). If published, this will include your full peer review and any attached files.

Reviewer #1: No

Reviewer #2: No

---

## [Author Response · Author response to Decision Letter 0]

12 Jan 2021

Please see the file Response_to_Reviewers.pdf for my response. Thank you!

---

## [Editor Report · Decision Letter 1]

19 Jan 2021

Information Retrieval Interfaces in Virtual Reality - A Scoping Review Focused on Current Generation Technology

PONE-D-20-32580R1

Dear Dr. Schleuβinger,

We’re pleased to inform you that your manuscript has been judged scientifically suitable for publication and will be formally accepted for publication once it meets all outstanding technical requirements.

Kind regards,

M. Usman Ashraf, Ph.D

Academic Editor

PLOS ONE

---

## [Editor Report · Acceptance letter]

27 Jan 2021

PONE-D-20-32580R1 

Information retrieval interfaces in virtual reality - A scoping review focused on current generation technology 

Dear Dr. Schleußinger:

I'm pleased to inform you that your manuscript has been deemed suitable for publication in PLOS ONE. Congratulations! Your manuscript is now with our production department. 

Kind regards, 

on behalf of

Dr. M. Usman Ashraf 

Academic Editor

PLOS ONE